# Brain-Derived Neurotrophic Factor Regulates Ishikawa Cell Proliferation through the TrkB-ERK1/2 Signaling Pathway

**DOI:** 10.3390/biom10121645

**Published:** 2020-12-08

**Authors:** Maosheng Cao, Qiaoge Niu, XinYu Xiang, Chenfeng Yuan, Tariq Iqbal, Yuwen Huang, Meng Tian, Zijiao Zhao, Chunjin Li, Xu Zhou

**Affiliations:** College of Animal Sciences, Jilin University, Changchun 130062, China; caoms18@mails.jlu.edu.cn (M.C.); niuqg18@mails.jlu.edu.cn (Q.N.); xiangxy9917@mails.jlu.edu.cn (X.X.); yuancf18@mails.jlu.edu.cn (C.Y.); iqbaltariq9917@mails.jlu.edu.cn (T.I.); huangyw18@mails.jlu.edu.cn (Y.H.); tianmeng18@mails.jlu.edu.cn (M.T.); zhaozj18@mails.jlu.edu.cn (Z.Z.)

**Keywords:** BDNF, Estradiol-17β, Ishikawa cell, TrkB-ERK1/2 signaling pathway

## Abstract

(1) Background: Endometrial regulation is a necessary condition for maintaining normal uterine physiology, which is driven by many growth factors. Growth factors produced in the endometrium are thought to be related to the proliferation of endometrial cells induced by estradiol-17β (E_2_). In this study, we found that E_2_ can induce the secretion of brain-derived neurotrophic factor (BDNF) in Ishikawa cells (the cells of an endometrial cell line). Furthermore, Ishikawa cells were used in exploring the regulatory role of BDNF in endometrial cells and to clarify the potential mechanism. (2) Methods: Ishikawa cells were treated with different concentrations of BDNF (100, 200, 300, 400, and 500 ng/mL). The mRNA expression levels of various proliferation-related genes were detected through quantitative reverse transcription polymerase chain reaction, and the expression of various proliferation-related genes was detected by knocking out BDNF or inhibiting the binding of BDNF to its receptor TrkB. The expression levels of various proliferation-related genes were detected by performing Western blotting on the TrkB-ERK1/2 signaling pathway. (3) Results: Exogenous BDNF promoted the growth of the Ishikawa cells, but the knocking down of BDNF or the inhibition of TrkB reduced their growth. Meanwhile, BDNF enhanced cell viability and increased the expression of proliferation-related genes, including cyclin D1 and cyclin E2. More importantly, the BDNF-induced proliferation of the Ishikawa cells involved the ERK1/2 signaling pathway. (4) Conclusions: The stimulating effect of exogenous E_2_ on the expression of BDNF in the uterus and the action of BDNF promoted the proliferation of the Ishikawa cells through the TrkB-ERK1/2 signal pathway.

## 1. Introduction

Estradiol-17β (E_2_) plays an important role in overall reproductive activities in mammals [1]. The predominant mechanism of E_2_ action is nuclear E_2_ receptor expression in E_2_ target organs [2]. E_2_-mediated biological effects are mediated primarily through its receptors ERα and ERβ [2]. The endometrium has receptor ERα, which is one of the E_2_ target tissues [3,4,5,6]. E_2_ is up-regulated during embryo implantation in humans and mice [7,8,9], and E_2_ can proliferate mouse uterine matrix, endometrial glandular epithelial cells, endometrial cells, and glandular epithelial cells [4,10,11]. Moreover, E_2_ and progesterone can stimulate endometrial stromal cell proliferation [3], which is necessary for establishing and maintaining pregnancy. The endometrium loss of the E_2_ receptor gene results in uterine dysplasia and female infertility [12]. However, high concentrations of E_2_ can induce endometrial cell proliferation, which may lead to endometrial cancer [5,13] or endometriosis [14].

During embryo implantation, the uterine cavity epithelium or glandular epithelium secretes some growth factors that promote embryo implantation and development through autocrine or paracrine mechanisms [15]. A high E_2_ level during embryo implantation may stimulate endometrial cells to secrete some growth factors, which are responsible for endometrial proliferation [16,17]. Therefore, identifying growth factors secreted by E_2_-regulated endometrial cells is necessary for deepening the understanding of the molecular signaling network of the uterus during pregnancy. In the current research, an in vitro simulated embryo attachment experiment was performed, in which an Ishikawa cell line was mainly used as the endometrial epithelial cell line [18,19]. The Ishikawa cell line is of great significance to the study of embryo implantation in vitro [20].

The brain-derived neurotrophic factor (BDNF) is a growth factor belonging to the neurotrophic family and plays an important role in the nervous system [21]. Mice lacking BDNF in their brains develop deficiencies and die early [22]. BDNF is essential for animal reproduction [23]. It is localized in different types of cells in the ovary, including oocytes, granulosa cells, and endometrial cells [24] and is one of the growth factors secreted by granulosa cells and endometrial cells during follicular development [25]. Exogenous BDNF can promote the growth of granulosa and cumulus cells [26]. Our previous studies revealed that BDNF is involved in the synthesis of steroid hormones [26]. In the uterus, BDNF and its receptor TrkB are expressed in the uterus luminal and glandular epithelia of humans, mice, rats, and sheep [27,28]. A normal endometrium produces weak levels of TrkB, whereas endometrial cancer tissues produces high TrkB levels. The up-regulation of TrkB promotes epithelial-mesenchymal transition in endometrial carcinoma [29]. BDNF is involved in the neurogenesis of endometrial stem cells and affects endometrial physiology [30,31]. Compared with non-pregnant animals, ewes have highly expressed BDNF in their uteri during pregnancy [28]. BDNF is detected at different embryonic stages [30,32].

A growing body of evidence has demonstrated the interaction between E_2_ and BDNF. Exogenous E_2_ can promote the expression of BDNF and its receptor TrkB in rat brain [33,34]. In ovariectomized mice, exogenous E_2_ increases the expression of BDNF in the uterus [35]. However, whether E_2_ can stimulate the secretion of BDNF in the endometrium as well as the underlying mechanism of the autocrine effect of BDNF in the endometrium remain unclear. Therefore, this study investigated the effect of BDNF on the proliferation of endometrial cells and its potential mechanism with a cell model.

## 2. Materials and Methods

### 2.1. Cell Culture and Treatments

Ishikawa cells from a human endometrial cancer cell line were provided by the European Collection of Animal Cell Cultures (ECACC 99040201). An RPMI/1640 medium (HyClone, Logan, UT) composed of 1% (vol/vol) penicillin/streptomycin (Invitrogen, CA, USA) and 10% inactivated fetal bovine serum (FBS, HyClone, Logan, UT) were used to culture the cell at 37 °C in 5% CO2/95% air environment. When the cell density was approximately 70–80%, the cells were treated with 100 nM E_2_ (Sigma-Aldrich, St. Louis, MO, USA) in a serum-free medium RPMI/1640 for 24 h. When cell density was about 70–80%, the cells were pretreated with 10 μM/mL PD98059 (Beyotime Biotechnology, Shanghai, China) and 100 ng/mL K252a (Invitrogen, Life Technologies, USA) for 30 min each. PD98059 is an inhibitor of mitogen-activated protein kinase (MAPK) activity, and K252a was used the inhibitor of the TrkB receptor. Then, the cells were treated with 200 ng/mL BDNF (Sigma-Aldrich, St. Louis, MO, USA) in a serum-free RPMI/1640 medium for 24 h. The cells were then collected for quantitative real-time polymerase chain reaction (qRT-PCR), and Western blot analysis. Each experiment was replicated three times.

### 2.2. Transfection with Small Interfering RNA

Small interfering RNA (siRNA) and si--NC were purchased from RiboBio (Guangzhou, China). The BDNF siRNA sequence was 5′--GCUCAGUCAAGUGCCUUU--3′ and 5′--AAAGGCACUUGACUACUGAGC--3′. Briefly, the Ishikawa cells were grown in a 12-well plate with 2 × 10^5^ cells in each well and were incubated until they reached 70–75% confluence. For the transfection of BDNF siRNA, FuGENE^®^ HD (Roche, Basel, Switzerland) was used at a concentration of 50 nM according to the manufacturer’s protocol. The cells were treated with siBDNF in serum--free medium for 36 h. Subsequently, the cells were collected for qRT-PCR and Western blot analysis. Each experiment was performed in three replicates.

### 2.3. RNA Extraction and Quantitative Real-Time Polymerase Chain Reaction Analysis

An RNA isolation system (Sigma-Aldrich, St. Louis, MO, USA) was used to extract total RNA and a Prime Script RT reagent kit with a gDNA removal kit (Takara Bio, Otsu, Japan) was used in obtaining cDNA. A 20 µL total reaction system containing 10 uL of SYBR (Takara Bio, Otsu, Japan), 0.5 µM forward primer, 0.5 µM reverse primer, 5 ng/µL cDNA, and DEPC water was used. A qRT-PCR machine was set at 95 °C for 2 min followed by 40 cycles of 95 °C for 15 s and 60 °C for 30 s. Data were analyzed using comparative 2^−△△Ct^ with normalization to the Gapdh housekeeping gene. The primers were synthesized in Kumei, Changchun, China (Table 1).

### 2.4. Western Blot Analysis

The cells were treated with PD98059, K252a, BDNF, siNC (50 nM), and siBDNF (50 nM). RIPA lysate buffer (Beyotime Biotechnology, Shanghai, China) supplemented with 1 mM PMSF (Beyotime Biotechnology, Shanghai, China) was used to lyse the cells for protein isolation. The sample was centrifuged at 12,000 g for 10 min at 4 °C, for the harvesting of cellular proteins. The concentrations of the proteins were measured using a BCA protein assay kit (Beyotime Biotechnology, Shanghai, China). The total proteins were separated with 12% SDS-PAGE, then transferred to a polyvinylidene difluoride (0.2 µm; Millipore, USA) membrane with a half-wet transfer system (Bio-Rad; Hercules, CA, USA). The membrane was blocked with 5% BSA (Sigma-Aldrich). After 1 h, the membranes were incubated with primary monoclonal antibodies specific to CCND1 (1:800, Bioworld), CCNE2 (1:800, Bio-world), BDNF (1:800, Bioworld), ERK1/2 (1:800, Bioworld), p-ERK1/2 (1:500, Bioworld) and GAPDH (1:8000, Bioworld) overnight at 4 °C. The membranes were washed four times with TBST. Each wash had an interval of 8 min. HRP-conjugated goat anti-rabbit secondary antibody (1:10,000, Bioworld, Minnesota, USA) were used in incubating the membranes for 1 h at room temperature. The membranes were washed again with TBST for four times with 8-min intervals. The proteins were detected using a BeyoECL star kit (Beyotime Biotechnology, Shanghai, China). A Tanon gel imaging system (Tanon) was used in analyzing the relative protein levels through grey scanning.

### 2.5. Cell Viability Assays

For the cell viability assay, Cell Counting Kit-8 (CCK-8, Beyotime Biotechnology, Shanghai, China) was used according to the protocol with the kit’s protocol. Ishikawa cells were planted in 96-well plates with approximately 1 × 10^4^ cells/well. After 12 h, the culture media was replaced with pre-warmed, fresh, serum-free medium pretreated with 0,100, 200, 300, 400, and 500 ng/mL BDNF successively for 24 h as indicated in the kit’s protocol. Another group of Ishikawa cells were planted in 96-well plates with approximately 1 × 10^4^ cells/well. After 12 h, the culture media were replaced with pre-warmed, fresh, and serum-free medium pretreated with siNC (50 nM) and siBDNF (50 nM) successively for 36 h as mentioned in the kit’s protocol. Then, each well was incubated with 10 μL CCK-8 solution for 2 h. The cells were counted using an ELX 800 universal microplate reader (BioTek, Highland Park, IL, USA), and absorbance was recorded at a wavelength of 450 nm (OD450). The reading performed in triplicate. Each experiment was performed in triplicate.

### 2.6. Colony Formation Assay

For colony formation, the cells were planted into six well plates with a rough density of approximately 1 × 10^4^ cells per well. The media (1% (vol/vol) penicillin/streptomycin, 2% FBS) was replaced every 2 days for 2 weeks. After 2 weeks, viable colonies were fixed with methanol and stained with crystal violet according to the protocol of (Zheng et al., 2017). The images were obtained with an inverted microscope (Olympus, Tokyo, Japan). Colonies comprising more than 50 cells were counted. Data were recorded and statistically analyzed.

### 2.7. Cellular Immunofluorescence

Ishikawa cells were cultured in a 12-well plate. When the cells reached 70% confluence, the cells were immersed in PBS three times for 3 min each time, and then the cells were immersed in ice-cold methanol for fixation for 20 min and rinsed with PBS three times for 3 min each time. Goat serum (10%) was then added and blocked for 30 min at room temperature. After blocking the 10% goat serum blocking solution was aspirated, without washing, then an appropriate amount of anti-TrkB (1:100; Bioworld, Nanjing, China) was added. The resulting solution was incubated overnight at 4 °C; The cells were then washed three times with PBS for 5 min each time and incubated with fluorescein (FITC) conjugated immunoglobulin G antibody (1:200; Bioworld, Nanjing, China). After incubation for 1 h at room temperature in the dark, the cells were washed three times with PBS for 5 min each time. Afterwards, the cells were incubated with DAPI solution (1:1000; Beyotime) for 10 min in the dark. Finally, the Ishikawa cells were observed using an Olympus fluorescence microscope (IX71; Olympus, Tokyo, Japan), and the Olympus application suite (cellSens Dimension; Olympus) was used in obtaining images. As shown in Figure 1, TrkB receptors were present in the Ishikawa cells.

### 2.8. Statistical Analysis

All the described experiments were carried out at least three times. All data were analyzed by one-way ANOVA and t-test with SPSS 19.0 software (Version X, IBM, Armonk, NY, USA). The difference between the control and treatment group was examined with Dunnett multiple- comparison test. Data were expressed as mean+ SD of three independent processes. A *p*-value of < 0.05 was considered significant, and a *p*-value of < 0.01 was considered to be a very significant difference.

## 3. Results

### 3.1. E_2_ promotes the Synthesis of BDNF Protein in Ishikawa Cells

Exogenous E_2_ increased the level of BDNF in the uteri of mice [35]. We speculate that estrogen promotes the expression of BDNF in Ishikawa cells. In the present study, the Ishikawa cells treated with E_2_ (100 nM, 1% DMSO) for 24 h, and the BDNF level was detected. The results showed that E_2_ increased BDNF level at the mRNA and protein levels (Figure 2A–C).

### 3.2. Effects of BDNF on Proliferation of Ishikawa Cells

The effect of estrogen on the proliferation of Ishikawa cells is mediated by ERRγ through AKT and ERK1/2 [36]. Our results showed that E_2_ promotes the synthesis of BDNF protein in Ishikawa endometrial cells. Therefore, we speculate that BDNF can promote Ishikawa cell proliferation. Cellular viability and colony formation assays were used in determining whether BDNF can promote the growth of Ishikawa cells. After treatment with different concentrations of BDNF for 24 h, BDNF increased the cell viability (Figure 3A) and proliferation (Figure 3B–C) of Ishikawa cells, especially when the BDNF concentration of 200 ng/mL. However, the knocking down of BDNF with siBDNF inhibited the viability (Figure 4D) and proliferation (Figure 4E–F) of the Ishikawa cells. To further determine BDNF-induced proliferation of Ishikawa cells, we evaluated the expression levels of proliferation-related genes (including CCND1 and CCNE2). The results showed that the knocking down of BDNF significantly regulated the expression of CCND1 and CCNE2 (Figure 4D–F). We further tested the ERK1/2 signaling pathway. The results showed that the level of p-ERK1/2 decreased after the knocking down BDNF treatment (Figure 4J), indicating that ERK1/2 participates in this process.

### 3.3. BDNF Promotes the Proliferation of Ishikawa Cells via TrkB

TrkB receptors of different types are present in the uterus. We used cell immunofluorescence to check the expression of TrkB in Ishikawa cells (Figure 1). To confirm whether TrkB receptors mediated the BDNF-induced proliferation of Ishikawa cells, we treated the cells with 200 ng/mL BDNF alone or with K252a (100 ng/mL, TrkB inhibitor, Invitrogen, Carlsbad, CA, USA) for 24 h. As shown in Figure 5A–E, K252a attenuated the viability and proliferation of the Ishikawa cells induced by BDNF. In addition, K252a decreased the expression of proliferation-related genes (Figure 5F–H). These results suggested that BDNF promotes cell proliferation mediated by TrkB.

### 3.4. BDNF Promotes Cell Proliferation through the MAPK/ERK1/2 Signaling Pathway

To explore the molecular mechanism of the BDNF-induced proliferation of Ishikawa cells further, we detected mitogen-activated protein kinase (MAPK)/ERK1/2 signaling pathway. The results showed that the level of p-ERK1/2 increased after BDNF treatment (Figure 6A–B), indicating that ERK1/2 participates in this process. However, PD98059 (10 µM, a specific inhibitor of MAPK, Beyotime) significantly decreased BDNF-induced cell proliferation and gene expression (Figure 6C–E). These results showed that BDNF promotes the proliferation of Ishikawa cells via the MAPK/ERK1/2 signaling pathway.

## 4. Discussion

E_2_ plays a vital role in the development and function of the female reproductive tract. The uterus responds to cyclical changes in estrogen and progesterone levels in preparation for embryo implantation. During embryo implantation in mice, estrogen in the uterus is up-regulated [7]. In mice, estrogen stimulates uterine epithelial cell proliferation by causing the nuclear accumulation of CCND1 and activation of cyclin E and cyclin A/CDK2 kinase complexes [37]. The effect of E_2_ on the proliferation of Ishikawa cells is mediated by ERRγ through AKT and ERK1/2 [36]. In breast cells, estrogen can regulate the expression and function of c-Myc and CCND1, and downstream E2F and CHD8 of CCND1 can activate CCNE2-Cd2, thereby promoting the transcription of CCNE2 and promoting the proliferation of breast epithelial cells [38,39]. Thus, the entry of CCND1 into the nucleus is the central regulatory point of estrogen action in the uterus. CCND1 and CCNE2 belong to the highly conserved cyclin family [40]. CCND1 and CCNE2 are cyclins that regulate the transition from the G1 phase to the S phase in the mitotic cell cycle and promote cell proliferation [26]. In this study, we demonstrated that these genes are involved in BDNF induced proliferation by Ishikawa cells.

BDNF plays important roles in reproduction [23] and embryo implantation. BDNF can promote mouse embryo development before implantation [23]. It can prepare an embryo for implantation and affect maternal placenta development by promoting the growth of uterine trophoblast cells [30]. Moreover, BDNF promotes the proliferation of porcine endometrial epithelial cells by activating the PI3K and MAPK signaling pathways [41]. After embryo implantation, the BDNF/TrkB signaling system can promote the growth of trophoblast cells through autocrine or paracrine actions and thereby promote the development of the placenta [30]. Exogenous estradiol increases the level of mRNA (more than sixfold) and protein (more than five-fold) of BDNF in the uteri of ovariectomized mice [35]. In the present study, estrogen-treated Ishikawa cells were found to promote BDNF gene expression. Ishikawa cells were used in assessing the effects of BDNF and TrkB kinase interruption on cell proliferation. K252a is an inhibitor of the tyrosine protein kinase activity of neurotrophin receptors [42]. We observed that apoptotic cells increased after K252a treatment. Thus, we considered that TrkB activated by BDNF plays a role in the survival of Ishikawa cells.

BDNF plays an important role in the reproductive system as a paracrine or autocrine signal, and extracellular signal-regulated protein kinase 1/2 (ERK1/2) signaling pathways can participate in the regulation of various cell functions, including cell adhesion, cell cycle progression, cell proliferation, and transcription [43]. Our previous studies showed that BDNF promotes the proliferation of bovine granulosa cells by increasing the expression of CCNA1 and CCNE2 through the BDNF-TrkB-ERK1/2 signal pathway [44]. BDNF can promote trophoblastic cell proliferation through the TrkB-ERK1/2 signal pathway [30]. Our present results showed that BDNF increases phosphorylated ERK1/2 levels and affects the proliferation of Ishikawa cells through the ERK1/2 signaling pathway. Therefore, we speculate that exogenous E_2_ stimulates Ishikawa cells to secrete BDNF and BDNF affects Ishikawa cell proliferation through the TrkB-ERK1/2 signaling pathway.

## 5. Conclusions

This study demonstrated the stimulating effect of exogenous E_2_ on the expression of BDNF in Ishikawa cells and the mechanism by which BDNF promotes the proliferation of Ishikawa cells through the TrkB-ERK1/2 signal pathway. These data provide evidence of the regulatory role of BDNF in uterine development, implying that BDNF plays an important role in uterine development.

## Figures and Tables

**Figure 1 biomolecules-10-01645-f001:**
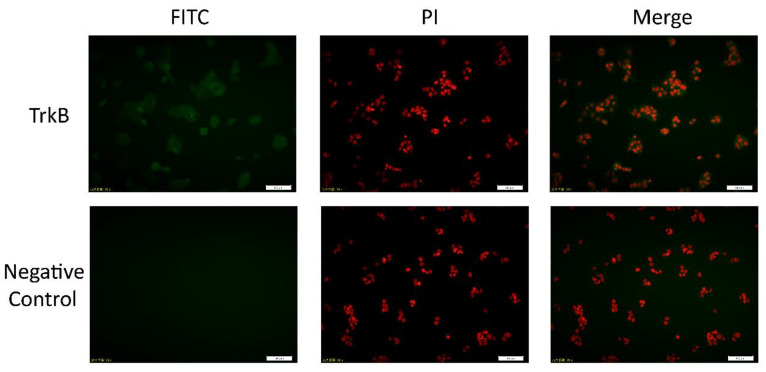
Determination of TrkB protein in Ishikawa cells by cell immunofluorescence. The Ishikawa cells were incubated with anti-TrkB or primary antibody dilution (negative control), then coupled with FITC-conjugated secondary antibody (FITC; green) and propidium iodide (PI; red) for the detection of nuclei. The picture (merge) was the result of the merging between the two fluorescences. Scale bar = 50 µm. TrkB: tyrosine kinase receptor β.

**Figure 2 biomolecules-10-01645-f002:**
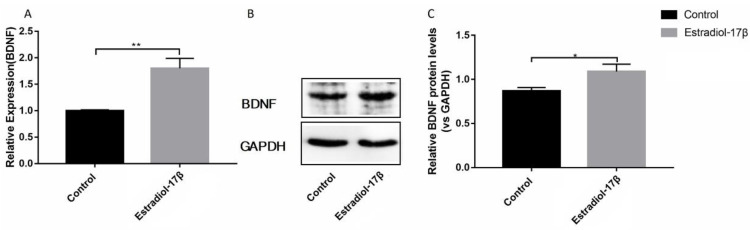
E_2_ up-regulated transcription/translation of BDNF in the Ishikawa cells. (**A**) Relative mRNA expression levels of BDNF gene were detected by qRT-PCR. (**B**) Protein levels of BDNF genes were detected by Western blot analysis. (**C**) Relative protein levels of BDNF genes were analyzed by grey scanning. The expression levels of mRNA and proteins were normalized by the GADPH gene, which plays an important role as an internal reference calibration for qRT-PCR and Western blot analysis. Statistical analysis is shown. * *p* < 0.05, ** *p* < 0.01.

**Figure 3 biomolecules-10-01645-f003:**
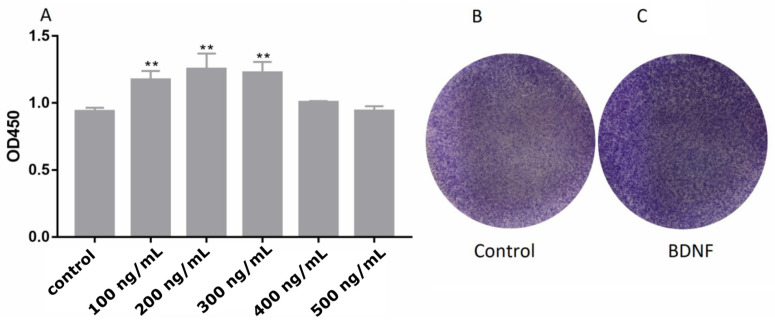
Brain-derived neurotrophic factor (BDNF) promoted Ishikawa cells’ viability and proliferation. (**A**) The viability of Ishikawa cells was measured by CCK-8 after treated with various concentrations of BDNF. BDNF significantly promoted Ishikawa cells viability at concentrations of 200 ng/mL. (**B**) Colony formation assay was used to evaluate cell proliferation in the control. (**C**) Colony formation assay was used to evaluate cell proliferation in the 200 ng/mL BDNF. Statistical analysis is shown. ** *p* < 0.01.

**Figure 4 biomolecules-10-01645-f004:**
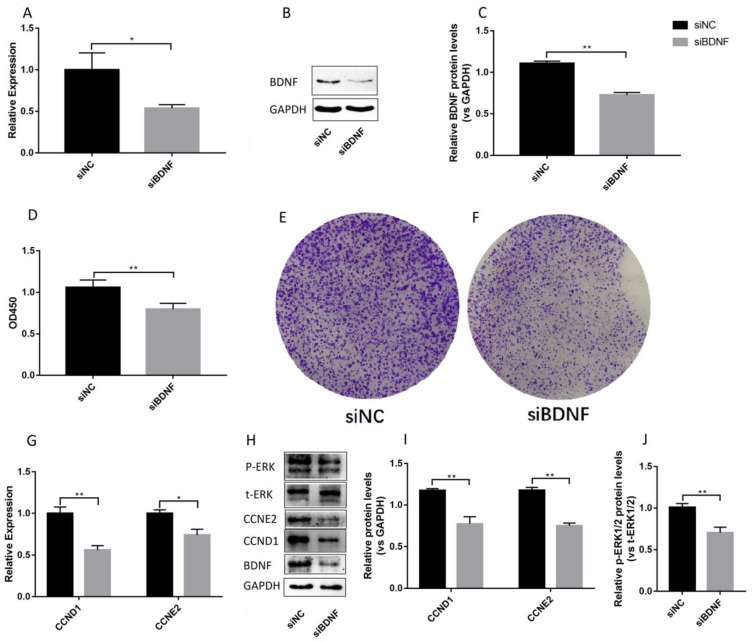
The effect of siBDNF on the proliferation of Ishikawa cells. The Ishikawa cells were transfected with siNC (control) or siBDNF for 36 h. qRT-PCR and Western blot analysis were used in detecting the expression levels of BDNF mRNA (**A**) and protein (**B**). (**C**) Relative protein levels were analyzed by grey scanning. (**D**) Cell counting kit-8 analysis was performed for the detection of cell viability. (**E**) Colony formation assay was used in detecting the proliferation of cells treated with siNC. (**F**) Colony formation assay was used in detecting the proliferation of cells treated with siBDNF. (**G**) Relative mRNA expression levels of CCND1 and CCNE2 genes were detected by qRT-PCR. (**H**) Protein levels of CCND1 and CCNE2 genes, P-ERK, and t-ERK were detected by Western blot analysis. (**I**) Relative protein levels of CCND1 and CCNE2 genes were analyzed by grey scanning. The expression levels of mRNA and proteins were normalized by the GADPH gene, which plays an important role as an internal reference calibration for qRT-PCR and Western blot analysis. (**J**) Relative protein levels of P-ERK and t-ERK were analyzed by grey scanning. Statistical analysis is shown. * *p* < 0.05, ** *p* < 0.01.

**Figure 5 biomolecules-10-01645-f005:**
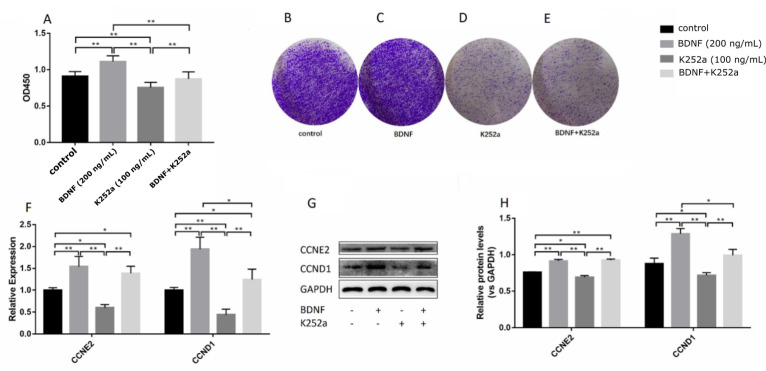
Effects of BDNF alone or that combined with K252a on the proliferation in Ishikawa cells. Ishikawa cells were monocultured with 100 ng/mL K252a for 30 min and then cocultured with 200 ng/mL BDNF for 24 h. (**A**) Cell counting kit-8 analysis was performed for the detection of cell viability. (**B**) Colony formation assay was used in detecting control cell proliferation. (**C**) Colony formation assay was used in detecting the cell proliferation treated with BDNF. (**D**) Colony formation assay was used in detecting the cell proliferation treated with K252a. (**E**) Colony formation assay was used in detecting the cell proliferation treated with K252a and BDNF. (**F**) Relative mRNA expression levels of CCND1 and CCNE2 genes were detected by qRT-PCR. (**G**) Protein levels of CCND1 and CCNE2 genes were detected by Western blot analysis. (**H**) Relative protein levels of CCND1 and CCNE2 genes were analyzed by grey scanning. The expression levels of mRNA and proteins were normalized by GADPH gene, which plays an important role as an internal reference calibration for q RT-PCR and Western blot analysis. Statistical analysis is shown. * *p* < 0.05, ** *p* < 0.01.

**Figure 6 biomolecules-10-01645-f006:**
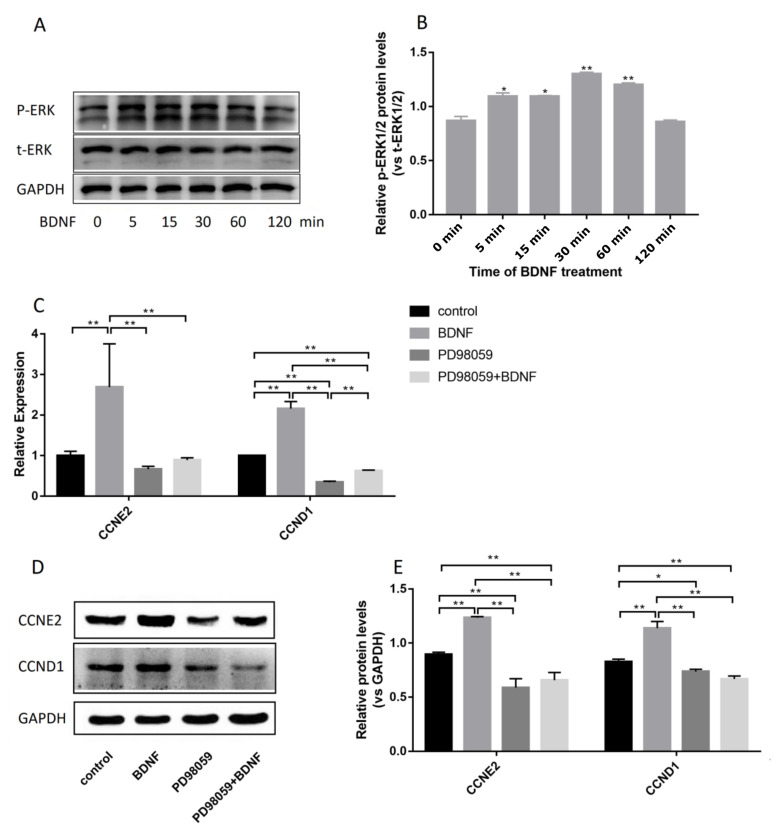
Effects of BDNF on proliferation of Ishikawa cells through TrkB–ERK1/2 signal transduction pathway. Ishikawa cells were monocultured with 10 μM/mL PD98059 for 30 min and then cocultured with 200 ng/mL BDNF for 24 h. (**A**) Ishikawa cells were treated with 200 ng/mL BDNF for 0, 5, 15, 30, 60, and 120 min. Protein levels of t-ERK and P-ERK were detected by Western blot analysis. (**B**) Relative protein levels were analyzed by grey scanning. (**C**) Relative mRNA expression levels of CCND1 and CCNE2 genes were detected by qRT-PCR. (**D**) Protein levels of CCND1 and CCNE2 genes were detected by Western blot analysis. (**E**) Relative protein levels of CCND1 and CCNE2 genes were analyzed by grey scanning. The expression levels of mRNA and proteins were normalized by the GADPH gene, which plays an important role as an internal reference calibration for qRT-PCR and Western blot analysis. Statistical analysis is shown. * *p* < 0.05, ** *p* < 0.01.

**Table 1 biomolecules-10-01645-t001:** Sequences of primers used in qRT-PCR.

Name	Sequences
*Gapdh*	F-5ACCCATCACCATCTTCCAGGAG3’
	R-5’GAAGGGGCGGAGATGATGAC3’
*Bdnf*	F-5’TCTGGAGAGCGTGAATGG3’
	R-5’AGGCACTTGACTACTGAGC3’
*CCNE2*	F-5’AAGAGGAAAACTACCCAGGATG3
	R-5’ATAATGCAAGGACTGATCCCC3’
*CCND1*	F-5’CCTCGGTGTCCTACTTCAAATG3’
	R-5’GCGGTCCAGGTAGTTCATG3’

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
