# Peer review of "Brain-Derived Neurotrophic Factor Regulates Ishikawa Cell Proliferation through the TrkB-ERK1/2 Signaling Pathway"

_biomolecules, 2020, doi:10.3390/biom10121645_

Round 1
Reviewer 1 Report
- There are many grammatical errors that make the manuscirpt difficult to read and understand.
- The order of the methods should match the order results are presented.
- There are not methods for all the results and figures.
- Why was the significant results of 100 ng/mL and 300 ng/mL BDNF not addressed?
- To further solidify the interaction of E2 and BDNF, experiments with E2 and TrkB inhibitor would be useful.
- The clinical significance is not clear. How can BDNF be used to treat uterine disorders?
Author Response
Manuscript ID: biomolecules-986034
Manuscript title: Brain-derived neurotrophic factor regulates Ishikawa cells proliferation through TrkB-ERK1/2 signaling pathway
Dear Editor:
We are grateful for your kind letter and reviewers’ constructive comments concerning our manuscript (Manuscript ID: biomolecules-986034). All the comments are highly insightful and enable us to greatly improve the quality of our manuscript. In the following pages are our point-by-point responses to the comments. Revisions in the text are marked for additions. We hope that you and the reviewers will be satisfied with the revisions and responses.
Yours sincerely,
Correspondence should be addressed to:
Chunjin Li and Xu Zhou
llcjj158@163.com and xzhou65@vip.sina.com

Reviewer 2 Report
This manuscript presents a mechanistic study about the estrogen-mediated regulation of the BDNF-TRKB signaling pathway in the endometrium, using Ishikawa cancer cells as a model. The experiments were performed in a logical manner, and the use of siRNAs, small molecule inhibitors, miscroscopy and other molecular techniques, support the authors’ conclusions about the E2-control of BDNF-TRKB signaling. However, in order to be suitable for publication, several structural issues in the paper’s presentation and description of the results must be addressed. These are outlined below as major and minor issues:
Major:
- Fig 1: The intensity of the TRKB staining is extremely low and poorly visible. Authors should obtain higher magnification images and display them as insets or a separate panel in the figure. Also, the reference to Figure 1 in the Results is not in a logical order and should be described first or the figures should be rearranged.
- What is the mechanism of action leading to E2-mediated BDNF activation? Do previously published ERalpha ChIPseq data show binding to the BDNF promoter? Does the BDNF promoter contain EREs, half EREs?
- Authors need to include more thorough descriptions of how the analyses were performed in the results section, i.e., for Figure 4, there is no reference to the time of transfection that the analyses were performed (is this 24, 48, or 72 hours post-transfection?).
- Figure 5, TRKB inhibitor (K252a) was used. The authors should include a brief description and reference about the mechanism of action of this inhibitor.
Minor:
- Thorough language/grammatical review, there are many mistakes, inconsistencies throughout the manuscript
- Ishikawa cells- inconsistently capitalized or not capitalized
- Clarify the statement in Line 38-39 about hyperplasia occurring during pregnancy
- Results section lines 225, 230: should rephrase the statements as hypotheses rather than question marks.
- Authors should use the conventional gene/protein nomenclature throughout the paper
- Each figure legend should state the number of technical and biological replicates that were performed for each experiment
- Each figure legend should contain a description of the statistical analysis used for those data
- There is no mention or reference to many of the figure panels in the Results section. Each result should be described in detail to convey the conclusions
- Line 263- do authors mean c-Myc?
Author Response

(The authors gave the same response as above.)
